# Neuroinflammation in Vascular Cognitive Impairment and Dementia: Current Evidence, Advances, and Prospects

**DOI:** 10.3390/ijms23116224

**Published:** 2022-06-02

**Authors:** Zhengming Tian, Xunming Ji, Jia Liu

**Affiliations:** 1Laboratory of Brain Disorders, Beijing Institute of Brain Disorders, Ministry of Science and Technology, Collaborative Innovation Center for Brain Disorders, Beijing Advanced Innovation Center for Big Data-Based Precision Medicine, Capital Medical University, Beijing 100069, China; tianzm01@163.com; 2Department of Neurosurgery, Xuanwu Hospital, Capital Medical University, Beijing 100069, China

**Keywords:** hypoxia, ischemia, microglia, neuroinflammation, vascular cognitive impairment and dementia

## Abstract

Vascular cognitive impairment and dementia (VCID) is a major heterogeneous brain disease caused by multiple factors, and it is the second most common type of dementia in the world. It is caused by long-term chronic low perfusion in the whole brain or local brain area, and it eventually develops into severe cognitive dysfunction syndrome. Because of the disease’s ambiguous classification and diagnostic criteria, there is no clear treatment strategy for VCID, and the association between cerebrovascular pathology and cognitive impairment is controversial. Neuroinflammation is an immunological cascade reaction mediated by glial cells in the central nervous system where innate immunity resides. Inflammatory reactions could be triggered by various damaging events, including hypoxia, ischemia, and infection. Long-term chronic hypoperfusion-induced ischemia and hypoxia can overactivate neuroinflammation, causing apoptosis, blood–brain barrier damage and other pathological changes, triggering or aggravating the occurrence and development of VCID. In this review, we will explore the mechanisms of neuroinflammation induced by ischemia and hypoxia caused by chronic hypoperfusion and emphasize the important role of neuroinflammation in the development of VCID from the perspective of immune cells, immune mediators and immune signaling pathways, so as to provide valuable ideas for the prevention and treatment of the disease.

## 1. Introduction

Vascular cognitive impairment and dementia (VCID) is a severe cognitive dysfunction syndrome caused by ischemia and hypoxia injury due to long-term chronic low perfusion in the whole brain or local brain region. Further, it belongs to a group of heterogeneous brain diseases caused by multiple factors, such as neuroinflammation and Tau pathology. These risk factors could induce different types and degrees of neurovascular injury, leading to the destruction of the nutritional coupling of neurovascular units, further aggravating the reduction in cerebral blood flow, thereby promoting the continuous decline in cognitive ability, and ultimately leading to cognitive impairment [1,2,3,4]. Moreover, VCID’s cognitive impairment is more severe than other neurodegenerative diseases such as Alzheimer’ s disease (AD) [1,5]. Major deficits in terms of attention, information processing and executive function are caused by frontal cortex circuit interruption induced by frequent subcortical vascular lesions [6]. With the exception of Alzheimer’s disease, VCID has become the world’s second most common type of dementia. The prevalence of VCID increases with age, reaching 1.5% in adults over 65 years of age [7]. Although the incidence of VCID is increasing every year, the cause remains unknown, making clinical classification, diagnosis and intervention of the disease difficult [8]. Ischemia and hypoxia caused by chronic cerebral hypoperfusion (CCH) are the fundamental factors that induce a series of pathological damage and promote the development of VCID. Understanding the molecular mechanism behind these pathological processes will provide a valuable theoretical basis for the prevention and treatment of VCID.

Neuroinflammation is a cascade of immune responses mediated by innate immune residents of the central nervous system (CNS)—glial cells (mainly microglia and astrocytes)—that can be triggered by damaging processes such as ischemia and hypoxia [9]. On the one hand, neuroinflammation can maintain the stability of the microenvironment, but it can also harm brain cells and neurons due to overactivation of inflammatory response [10,11]. An overactivated inflammatory response can lead to neuronal damage and even death, and destroy neurological function. Studies have shown that neuroinflammation plays a key role in the pathogenesis of various neurological diseases, such as cerebrovascular disease, Alzheimer’s disease, multiple sclerosis and Parkinson’s disease [10,12,13,14]. Ischemia and hypoxia caused by CCH can induce neuroinflammation. Excessive neuroinflammation is a major initiating and promoting factor in the onset and development of VCID. Identifying the underlying important role and mechanism will provide a reliable target for the intervention of VCID [15,16].

In this review, we describe the main etiology and existing pathogenesis of VCID, discuss the mechanism of ischemia- and hypoxia-induced neuroinflammation caused by chronic hypoperfusion, and highlight the important role of neuroinflammation in promoting the development of VCID in immune cells, immune mediators and immune signaling pathways, with the aim of providing potential directions and targets for the research and prevention of VCID.

## 2. Vascular Cognitive Impairment and Dementia

VCID is a heterogeneous brain disease caused by multiple factors. Persistent chronic cerebral hypoperfusion will lead to the persistence of an ischemic and hypoxic microenvironment in the brain. Ischemia and hypoxia will then induce a series of pathological reactions, such as neuroinflammation, oxidative stress, neurotrophic uncoupling and blood–brain barrier (BBB) destruction, and eventually lead to cognitive dysfunction.

### 2.1. Etiological Study of VCID

VCID is a severe cognitive dysfunction syndrome caused by continuous chronic cerebral hypoperfusion and can be subdivided into four main subtypes according to the Vascular Impairment of Cognition Classification Consensus Study (VICCCS) guidelines: post-stroke dementia, subcortical ischemic VCID, multiple infarction (cortical) dementia, and mixed dementia [17,18]. As these four subtypes have the common phenomenon of a dysfunctional cerebral vascular regulation mechanism and insufficient cerebral blood flow (CBF), ischemia and hypoxia have become the main characteristics and main pathogenic factors of VCID. In the brain of VCID, there is an environment with ischemia and hypoxia, which induces different types and degrees of neurovascular injury, leading to the destruction of the nutritional coupling of neurovascular units (NVU), further aggravating the decrease in cerebral blood flow, promoting the continuous decline in cognitive ability, and eventually leading to dementia [19].

The central nervous system is known to be extremely sensitive to blood supply, and normal blood supply to neurons plays a significant role in the occurrence of neural activities [20]. The blood supply of the CNS can simply be blocked by various vascular risk factors, leading to cerebral ischemia [21]. Cerebral ischemic disease is a common form of stroke that has become one of the leading causes of high morbidity and mortality worldwide. It is primarily the result of vascular occlusion caused by thrombosis or embolism [22,23]. It is characterized by insufficient blood supply to the brain, resulting in insufficient oxygen and nutrient transport, triggering lesions that cause increased permeability of the BBB and inflammatory responses, leading to apoptosis directly or indirectly [24].

The function and vitality of neurons, as well as the structural and functional integrity of the brain, are significantly reliant on the stable supply of oxygen and glucose. Therefore, the maintenance of an appropriate oxygen concentration is of great significance for the normal functioning of CNS [25]. With the evolution of the oxygen-sensing and regulation system, the regulation of oxygen homeostasis in the brain has become very mature. For example, by adjusting the vascular tension, the local blood flow supply and oxygen supply can be increased, which increases the oxygen supply in the brain area with the change in metabolic demand, and provides the basis for the enhancement of nerve activity [26]. However, when oxygen supply is insufficient due to various factors, such as under hypoxic conditions, neurons will not be able to perform normal functions, and can even lead to cell death [27]. Hypoxia is a common pathophysiological mechanism that can lead to systemic pathological reactions. In recent years, the effect of hypoxia on the central nervous system has attracted more and more attention. Studies have found that hypoxia is related to many central nervous system diseases, including stroke, head trauma and neurodegenerative diseases [28] (Figure 1).

The tight junction between vascular endothelial cells will weaken as cerebral blood flow decreases gradually in all or some of the brain regions. Astrocyte end foot is destroyed, the AQP4 water channel reduces, the ability of the pericytes to regulate vasomotor activity decreases, along with microglia activation and proliferation. Eventually, a series of pathological reactions are induced, leading to cognitive impairment.

### 2.2. Pathogenesis of VCID

#### 2.2.1. Neuroinflammation

Neuroinflammation is a cascade immune response mediated by innate immune-resident, CNS glial cells. The inflammatory biomarker levels in plasma and cerebrospinal fluid (CSF) have been found to be altered in patients with VCID, indicating that neuroinflammation is involved in the pathogenesis of VCID [29]. In addition, in a VCID animal model of chronic cerebral hypoperfusion, vascular risk factors caused by cerebral ischemia and hypoxia can induce the immune cascade reaction, which can lead to white matter damage and neuronal loss, followed by learning and memory dysfunction [30], which suggests that neuroinflammation may be an important pathogenic factor of VCID. In VCID, the pathological changes in the cerebral vasculature will lead to ischemia and hypoxia, and then lead to excessive neuroinflammation. In this process, immune cells will be activated, and inflammatory mediators and inflammatory pathways will be regulated [31,32]. Such neuroinflammation, on the one hand, can induce neuronal dysfunction and even death, leading to cognitive dysfunction; on the other hand, it will continue to damage the vasculature, induce more serious vascular pathological changes and dysfunction, and eventually lead to more serious cognitive dysfunction and even dementia [33,34].

#### 2.2.2. Oxidative Stress

When reactive oxygen species (ROS) are produced in excess and the cellular antioxidant system is unable to regulate them properly, oxidative stress occurs, resulting in the destruction and damage of cellular functions, and occasionally cellular death [35]. Studies have shown that oxidative stress plays a crucial role in the pathogenesis of VCID patients and animal models. Increased levels of ROS in blood vessels and decreased bioavailability of nitric oxide were found in VCID patients, leading to endothelial dysfunction in vascular diseases and eventually inducing VCID [36]. Markers of oxidative stress (isoprene) and inflammatory response (cytokines and adhesion molecules) were up-regulated in damaged white matter in the CCH model [37]. Numerous enzymatic and non-enzymatic antioxidants were found to be decreased during chronic hypoperfusion [38]. It is suggested that vascular risk factors induced by ischemia and hypoxia can lead to oxidative stress and may induce VCID.

#### 2.2.3. BBB Dysfunction

BBB is a multicellular vascular structure, which separates the CNS from the peripheral blood circulation and controls the passage of molecules and ions, while transporting substances according to neuronal needs, so as to protect the brain from toxins and pathogens [39,40]. BBB refers to the barrier between plasma and brain cells formed by cerebral capillary wall and glial cells, and the barrier between plasma and cerebrospinal fluid formed by choroid plexus. These barriers can prevent certain substances (mostly harmful) from entering the brain from the blood [41,42]. Increased plasma protein albumin levels in the cerebrospinal fluid of patients with vascular cognitive impairment (VCI) indicate that the BBB’s permeability has shifted and normal barrier function has been compromised [43]. Plasma proteins, including complements, fibrinogen, albumin and immunoglobulin, were also detected in astrocytes within white matter lesions [44]. In addition to BBB disruption in patients with VCID, BBB permeability also increased in animal models with prolonged hypoperfusion [45]. These findings suggest that ischemia–hypoxia-induced vascular risk factors increase BBB permeability and induce VCID.

#### 2.2.4. Neurovascular Nutrient Uncoupling

The neurovascular unit (NVU) refers to the integration of interactions between multiple cells in the brain, which plays a vital role in brain development, maintenance and function [46]. This concept was first proposed in July 2001 to emphasize the unique relationship between brain cells and cerebral vessels. A complete NVU comprises neurons, astrocytes, microglia, vascular endothelial cells, perivascular cells, basement membrane and extracellular matrix [47]. Under normal physiological conditions, neurons, astrocytes, oligodendrocytes and vascular and perivascular cells are nutritionally and metabolically interdependent and play a decisive role in brain development, function and response to injury [48]. However, under long-term hypoperfusion, the cellular signal transduction and nutritional coupling between neurovascular units become abnormal, which causes vascular injury, and induces vascular risk factors, which then causes neuronal injury, glial cell activation and proliferation, vascular endothelial cell metabolism disorders, pericytes loss or dysfunction, extracellular matrix degradation, etc., and the overall damage to NVU function [49]. Impaired NVU induces a series of cascade reactions in the body, such as BBB damage, oxidative stress and neuroinflammation, and induces the formation of VCID [48].

#### 2.2.5. Others

In addition to the above mechanisms, there are still several theories on the pathogenic mechanism of VCID. For example, in the CCH model, due to the continuous decrease in CBF, mitochondrial morphology and function are damaged, which eventually leads to neuronal damage and cognitive dysfunction [50,51]. CCH leads to myelin sheath damage and demyelination. The loss of myelin will slow down the propagation of axon potential and lead to axon loss, which eventually leads to brain dysfunction [52,53]. In addition, the neurotoxicity induced by excessive zinc ions (Zn^2+^) released after CCH induces nerve degeneration and promotes the incidence and development of VCID [54].

## 3. Relationship between the Etiology of VCID and Neuroinflammation

Neuroinflammation plays an important role in many neurodegenerative diseases. Ischemia and hypoxia caused by chronic cerebral hypoperfusion can induce neuroinflammation, which can lead to neuronal dysfunction and even death, resulting in cognitive dysfunction.

### 3.1. Neuroinflammation

Neuroinflammation refers to a series of immune cascades after injury signals are recognized by immune cells in the CNS [55]. It mainly includes the following four characteristics: microglia activation, increased cytokine and chemokine levels, recruitment of peripheral immune cells and local tissue damage [56,57]. During inflammation, immunogen molecules can activate microglia, leading to their transformation into ameboid morphology, and significantly increase the production of cytokines and ROS, inducing downstream immune response and oxidative stress [58]. Neuroinflammation can lead to white matter damage and neuronal loss, followed by learning and memory dysfunction, leading to and accelerating neurodegenerative diseases, which plays a central role in the early development of chronic diseases such as dementia [59].

### 3.2. Neuroinflammation Induced by Ischemia

In cerebral ischemic diseases, blood flow stagnation will lead to inflammatory reactions. When inflammation is activated, glial cells, neutrophils, monocytes and lymphocytes are activated, as are pro-inflammatory cytokines and inflammatory-related metabolites, promoting the continuous occurrence of inflammation, ultimately causing persistent tissue damage [60,61,62]. This inflammatory response can be divided into four periods [63,64,65,66]: (1) During the acute phase, microglia or macrophages remove necrotic and fragmented cells; (2) During the subacute stage, oxidative stress occurs, which promotes the generation of inflammatory mediators such as cytokines and adhesion molecules, and continuously induces inflammatory response; (3) During the chronic phase, inflammatory mediators and reactive oxygen species lead to mitochondrial and DNA damage, triggering cell necrosis and apoptosis for several days; (4) During the recovery period, with the decrease in inflammatory response level, various nutritional factors will be released to supplement cell damage caused by apoptosis. This kind of neuroinflammation cascade reaction will lead to apoptosis, BBB damage, and brain edema. Subsequent ischemia–reperfusion will promote oxidative stress response and increase the release of pro-inflammatory cytokines, triggering a series of pathological cascade reactions, directly or indirectly leading to pathological changes such as apoptosis and BBB damage [67]. In cerebral ischemia diseases such as stroke, neuroinflammation can significantly affect the normal function of neurons and cause neuronal damage [16].

### 3.3. Neuroinflammation Induced by Hypoxia

In hypoxic conditions, most clinical and experimental data suggest that neuroinflammation has dual effects. For example, it can remove dead cells and necrotic cell debris during immune activation. On the contrary, excessive inflammatory responses are harmful, leading to increased infarction and decreased neuroplasticity [68]. Therefore, excessive neuroinflammation under hypoxic conditions may be an important cause of cognitive dysfunction in VCID. Hypoxia induces the up-regulation of hypoxia-inducible factor-1α (HIF-1α), an oxygen-dependent nuclear transcription factor, and subsequently activates many cytokines, such as inflammatory transcription factor NF-κB, while promoting the expression of inflammatory and metabolic genes. Hypoxia induces the synthesis and release of inflammatory mediators and enzymes, and induces inflammatory responses, leading to tissue damage and neurological deficits in CNS [15,69,70,71,72]. In addition, hypoxia can also change the expression of miRNA, which can regulate inflammation by affecting microglial activation, or regulate the production of inflammatory cytokines either directly or indirectly [73,74].

## 4. Research Progress of Neuroinflammation in VCID

It is now widely accepted that excessive neuroinflammation in the CNS causes neurodegenerative disorders. In VCID, the continuous decrease in CBF causes cell death, and the subsequent release of cell debris will induce the neuroinflammation–immune cascade reaction. During this process, immune cells will be activated, and inflammatory mediators and pathways will be regulated to promote the development of cognitive dysfunction in VCID.

### 4.1. Immune Cells in VCID

Innate immunity is essential for maintaining normal brain function. Various types of cells, including neurons, microglia and astrocytes, are known to form a complex and rigorous system in the brain that works together to maintain the immune homeostasis of the brain and ensure timely removal of pathogens or endogenous abnormal proteins and cells that invade the brain [75]. In the CNS, astrocytes account for 19–40% of the glial cells, microglia account for 10%, oligodendrocytes account for 45–75%, and NG2 cells make up the remainder [76]. NG2 cells, the fourth kind of glial cells in the brain except astrocytes, microglia and oligodendrocytes, are considered to be precursor cells of oligodendrocytes [77]. Most NG2 cells in the adult brain remain static under physiological conditions [78]. However, the exact role and mechanism of NG2 cells in cerebral nerve immunity remain unclear. Recent studies have shown that glial activation occurs in chronic hypoperfusion models, which then results in neuroinflammation, regulation of inflammatory pathways, white matter damage, and loss of neurons in important brain regions such as the corpus callosum and hippocampus, leading to cognitive impairment [31,32,33].

Microglia are innate immune effector cells that play an important role in the physiological processes in the central nervous system. They are a group of phagocytes primarily responsible for clearing debris from CNS during development and disease [79,80]. NVU microglia bodies are highly ramified and constantly moving in the physiological state to monitor the CNS’s microenvironment for damage or infection indicators in order to respond and maintain brain homeostasis [81]. Microglia are rapidly activated when the microenvironment is damaged and infected. Microglial fragments and dead neurons are removed by phagocytosis and inflammatory cascades are activated [82]. In CCH models, neuroinflammation caused by chronic cerebral hypoperfusion induces white matter damage, while microglia are activated and transform themselves into other phenotypes [44,83]. Studies have shown that the expression of MHC-I/II and matrix metalloproteinase-2 (MMP-2) in the white matter microglia increased at day 3 after hypoperfusion, indicating early activation of microglia due to hypoperfusion [84]. Some microglia turn into protective phenotypes that secrete growth factors and promote tissue recovery after identifying damage markers in the microenvironment, then immediately trigger anti-inflammatory response to promote tissue repair [81,85]. In addition, in this model, protective microglia will engulf a large number of myelin fragments in the microenvironment, thereby maintaining stability [86]. Another part of microglia will turn into a pro-inflammatory phenotype that aggravates tissue damage [85]. Cytokines released by harmful microglia induced by CCH increase the release of proinflammatory cytokines, thereby aggravating neuroinflammation [86].

Astrocytes are a very important class of cells in the central nervous system that regulate multiple processes in the nervous system in both healthy and diseased states [16]. During development, astrocytes originate from the neural progenitor cells (NPC) in the subventricular zone (SVZ) and migrate along the radial glial process to fill the whole brain, which then differentiates in the target area, becoming a highly functional specialized regional definition subgroup [87]. Under physiological conditions, astrocytes stretch their end feet, encircle capillaries, promote the formation and maintenance of BBB, and separate the peripheral blood circulation from the highly controlled central nervous system microenvironment [88,89]. Further, they actively participate in the process of cell signal transduction and secrete neurotrophic factors to regulate synaptic development, neuronal differentiation and neuronal survival [87,90,91]. In synapses, astrocytes actively modulate synaptic transmission by releasing and clearing neurotransmitters and regulating extracellular ion concentrations [92,93]. Studies have used Hyperhomocysteinaemia (HHcy) to mimic VCID, leading to neuroinflammation, cognitive impairment, and disruption to BBB permeability [94]. Further, morphological and biochemical evaluation of astrocytes indicated that the terminal foot was broken. The labeling of the AQP4 channel was reduced, the protein/mRNA level of the potassium ion channel was reduced, and the cognitive disorder worsened with time [95].

Oligodendrocytes are myelinated cells in the CNS produced by oligodendrocyte progenitor cells (OPCs) [96]. Oligodendrocytes are distributed in the central nervous system and are smaller than astrocytes. With shorter and fewer processes, their nuclei are round, small and dense. The electron density of the cytoplasm is high under the electron microscope. They mainly contain mitochondria, nucleoprotein bodies and microtubules [97]. Normally, mature and differentiated oligodendrocytes produce myelinated lipoproteins. Myelin is an extended membrane tightly wrapped around axons [98]. Oligodendrocytes surround the axons to form an insulated myelin sheath that assists in the efficient jump-through of electrical signals while maintaining and protecting the function of neurons [99]. Studies have shown that CCH-induced white matter injury will mediate the differentiation disorder of OPCs, reduce the number of oligodendrocytes, and demyelinate the neurons, thus impairing the cognitive function [98].

Macrophages in the central nervous system include boundary-associated macrophages (BAMs) in addition to microglia. Compared with microglia, nonparenchymal brain macrophages (BAMs) have less individual development, function, and transcriptome characteristics during embryogenesis or adulthood. BAMs are mostly located in the nonparenchymal boundary of the central nervous system, such as the choroid plexus, meningeal, and perivascular space [100]. Macrophages of this type also have similar monitoring and phagocytosis functions as microglia and can also present antigens to lymphocytes [101,102]. At present, there are few studies on the function of BAMs under pathological conditions, but it is clear that they are involved in the pathogenesis of several types of brain diseases. For example, BAMs are involved in clearing Aβ deposition in AD models and mediating Aβ-induced oxidative stress [103,104,105]. In addition, since BAMs are located at the junction between blood vessels, brain parenchyma, and the immune system, they may play an important role in the pathogenesis of cerebrovascular diseases. For example, in the acute phase of ischemic stroke, BAMs proliferate, migrate, and acquire a proinflammatory phenotype, which results in increased vascular permeability, granulocyte recruitment, and neurological dysfunction [106,107]. Therefore, we consider that BAMs may also play an important role in the pathogenesis of VCID, which provides a new concept for the pathological study of VCID.

### 4.2. Inflammatory Mediators in VCID

Inflammatory mediators are important factors that mediate the occurrence and development of inflammatory response. Some inflammatory mediators have been shown to play a key role in inflammation during CCH [36]. Increased production of pro-inflammatory cytokines such as interleukin-1β (IL-1β), IL-6, and tumor necrosis factor-α (TNF-α) in the CCH model aggravates ischemic injury with the synergistic effect of oxidative stress [51,108]. Currently, studies have confirmed that inflammatory cytokines and chemokines were detected in the CCH model in both the acute and chronic phase. The production of IL-1β, IL-6 and TNF-α in the ischemic hemisphere increased significantly during the acute phase [109]. Increased MCP-1, IL-6 and TNF-α were also detected with a significant increase in anti-inflammatory cytokines IL-4 and IL-10 [110,111]. In the chronic stage, IL-1β, IL-6 and TNF-α were up-regulated in the hippocampus and white matter [110,112]. ICAM-1 and VCAM-1 were up-regulated in cerebral endothelial cells of CCH models, and the inhibition of ICAM-1 and VCAM-1 may protect cognitive impairment induced by chronic hypoperfusion [113,114]. Increased levels of matrix metalloproteinase (MMP) family inflammatory mediators were also detected in the cerebrospinal fluid of VCID patients [115]. Increased MMP-2 during CCH contributes to angiogenesis, but also induces BBB disruption. Inhibition of MMP-2 reduces BBB disruption, glial activation and white matter lesions [116]. Additionally, upregulation of MMP-9 results in vascular injury and demyelination [34].

### 4.3. Inflammatory Pathways in VCID

Current studies have shown that multiple inflammatory pathways are activated simultaneously in VCID, and their synergistic effect plays an important role in subsequent cellular signal transduction. In the CCH model, the Janus kinase (JAK)/signal transducer and activator of transcription (STAT) signaling pathway is rapidly activated as the integrity of the axon–glia structure is disrupted [117,118,119]. With the activation of glial cells, the secretion of glial-derived inflammatory cytokines increased rapidly, and the signal transduction of Toll-like receptor 4 (TLR4)/myeloid differentiation factor 88 (MyD88) was activated. The downstream apoptotic signal transduction p38/mitogen-activated protein kinase (MAPK) was also activated, and then the downstream cascade reaction was induced [112,120,121]. NF-κB and STAT3 are also activated in the chronic hypoperfusion model, leading to neuroinflammation in the capillary and hippocampus [114,122,123]. In addition, the C3-C3aR/ITGAM pathway is also activated after white matter injury [86] (Figure 2 and Table 1).

In VCID, with the increase in chronic hypoperfusion time, astrocyte activation, pseudopodia fracture, and AQP4 water channel decrease take place. Simultaneously, with microglia activation and phenotype transformation, phagocytosis of cell debris in the microenvironment occurs. The number of oligodendrocytes also decreases and demyelination occurs. Further, the inflammatory mediators IL-1β, IL-6, TNF-α, ICAM-1 and VCAM-1, and MMP-2 and MMP-9 in ischemic sites are up-regulated. In addition, JAK/STAT, TLR4/MyD88/p38/MAPK and NF-κB/STAT3 in the inflammatory pathway are also up-regulated, indicating that the neuroinflammatory response was enhanced. The sign “↑” in Figure 2 and Table 1 means up-regulation, and “↓” means down-regulation.

## 5. Relationship between AD and VCID

A number of pathological studies of dementia have shown that most patients have cognitive dysfunction caused by various factors, and about one-third have vascular lesions [125,126]. These findings indicate that VCID not only has a high incidence, but also coexists with other neurodegenerative diseases, such as AD [17]. Existing studies have shown that changes in the brain and systemic blood vessels usually coexist with AD pathology. For example, in most AD patients, vascular injuries to varying degrees are found, including vascular dysfunction, cerebral infarction, atherosclerosis, and cerebral amyloid vascular diseases [127,128,129,130,131]. The resulting cerebrovascular insufficiency may lead to the occurrence and development of cognitive dysfunction, and controlling vascular risk factors can reduce the cognitive dysfunction of AD, indicating that VCID is closely related to AD [127].

Intracranial atherosclerosis and cerebral amyloidosis are associated with cognitive dysfunction in AD patients [127,132]. In patients with AD, atherosclerosis in the Willis circular artery at the bottom of the brain leads to a decrease in oxygen and energy supply in the brain due to insufficient CBF perfusion, and eventually leads to cognitive dysfunction [133]. At the same time, low perfusion in this case promotes the production and accumulation of Aβ, and further aggravates atherosclerosis through nerve inflammation, oxidative stress, and endothelial dysfunction, forming a vicious circle of “atherosclerosis-chronic cerebral hypoperfusion-cognitive dysfunction” [134,135]. In addition, when there is a large accumulation of Aβ in cerebral vessels due to the damage of Aβ clearance ability, Aβ becomes toxic, which damages vascular endothelial cells, destroys the normal structure and function of the BBB, and continuously reduces CBF, which eventually leads to cognitive dysfunction [48,129]. At the same time, the occurrence of vascular pathology has also been also found in AD models. For example, in a pathological study in AD spontaneous genetic model (SAMP8 mice), severe endothelial dysfunction and vascular remodeling disease were evident, which greatly damaged cognitive dysfunction [136]. In addition, in APP_Sw/0_ transgenic mice, the pathological changes in the cerebral capillaries and cerebral blood flow were also evident [137,138]. Taken together, the results confirm that vascular pathological changes play a very important role in the development of AD.

A series of vascular pathological changes found in AD have been shown to produce a series of pathological reactions that cause chronic cerebral hypoperfusion. According to the former, this type of vascular injury induces an excessive neuroinflammatory response, and ultimately results in continuing and deteriorating cognitive dysfunction in AD patients. Therefore, considering the cognitive dysfunction induced by vascular pathological changes from the perspective of neuroinflammation is not only of great significance in the study of the new pathogenic mechanisms of VCID, but also provides a new concept for the study of the molecular mechanism of vascular injury, which may be applicable to other neurodegenerative diseases.

## 6. Relationship between Other Tauopathic Syndromes and VCID

Tauopathies, a group of neurodegenerative diseases characterized by abnormal microtubule-associated protein tau deposition in brain cells, are severe and untreatable [139,140]. Tau is a kind of microtubule-associated protein in the form of monomer in its natural state, which plays an important role in microtubule assembly and stability [4,141]. The pathological transition of tau from its soluble and monomeric state to its hyperphosphorylated, insoluble and filamentous state will lead to tau dysfunction and occur in many neurodegenerative diseases, such as Alzheimer Disease (AD), Progressive Supranuclear Palsy (PSP), Corticobasal Syndrome (CBS) and so on [2,3,142]. Current studies have shown that vascular risk factors have become an important pathogenesis of tau disease. On the one hand, hypertension or cerebral ischemia, oxidative stress and other vascular risk factors will induce pathological changes in tau, resulting in motor disorders [3,143]. For example, the P301L-tau Tg mouse model is one of the best characterized mouse models of tauopathy, and the motor disturbances developed by them have been correlated with the presence of hyperphosphorylated tau [144,145,146]. On the other hand, vascular risk factors can lead to cognitive dysfunction by inducing neurological damage. For example, diabetes has become the main risk factor for PSP, and the decrease in cerebral perfusion in PSP may be regulated by blood glucose variability, thereby inducing the occurrence and development of cognitive dysfunction [147,148]. In addition, diabetes can also be an important risk factor that participates in the cognitive decline process of other neurodegenerative diseases via mechanisms that involve cerebrovascular pathologies, such as AD [149,150,151].

In VICD, long-term CCH has become the main feature and important inducement of the disease. Studies have shown that CCH will induce tau hyperphosphorylation, and tau hyperphosphorylation seems to be very sensitive to cerebral hypoperfusion [152,153]. In addition, studies have shown that there is a general correlation between tau pathology and microglia activation. For example, when neurons overexpress tau and abnormally accumulate phosphorylated tau in mice, this will induce microglia activation and promote the occurrence of neuroinflammation [154,155]. In summary, the pathological changes and accumulation of tau induced by CCH in VCID will induce excessive neuroinflammation, which will lead to neurological dysfunction and aggravate cognitive dysfunction. Therefore, although no study has linked “Vascular pathogenesis-Tau pathology-Neuroinflammation-Cognitive impairment” in VCID yet, this might become an important pathogenic mechanism of VCID.

## 7. Conclusions

Since there is no clear disease classification and diagnostic criteria for VCID, the relationship between cerebrovascular pathology and cognitive impairment is still controversial. Further, the research on pathological mechanisms is not yet developed, so there is no suitable disease model for the continuity of scientific research. Therefore, most of the existing studies focus on the changes observed in immune cells among VCID models. Additionally, the research on inflammatory mediators and inflammatory pathways is not comprehensive enough, hence it is impossible to further explore whether neuroinflammation can really promote the occurrence and development of cognitive impairment. Since neuroinflammation occurs primarily in pathological conditions such as ischemia and hypoxia, which is consistent with the main pathogenesis of VCID, future research needs to focus on the mechanism of neuroinflammation in the development of cognitive impairment in VCID.

## Figures and Tables

**Figure 1 ijms-23-06224-f001:**
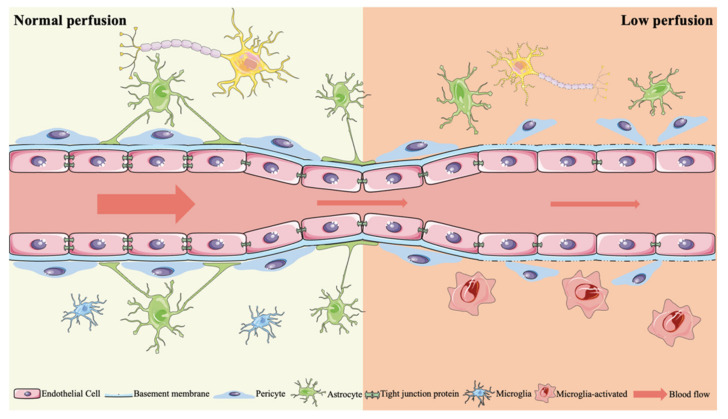
VCID induced by chronic cerebral hypoperfusion.

**Figure 2 ijms-23-06224-f002:**
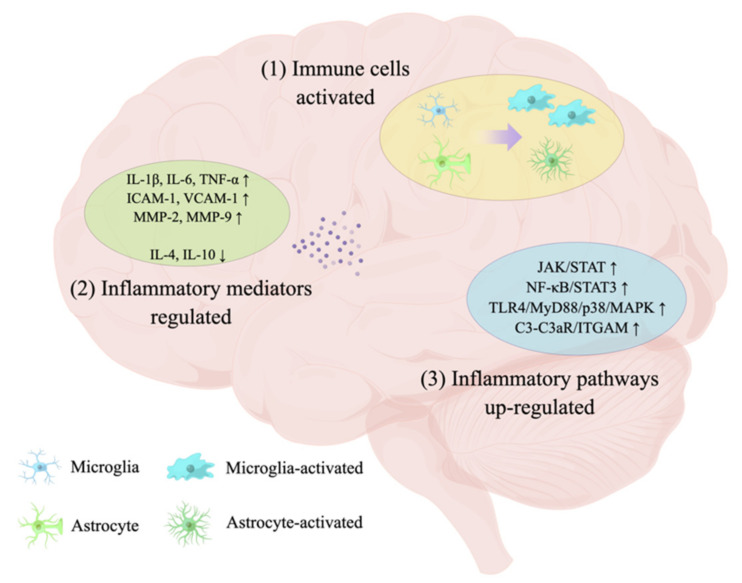
Neuroinflammation in VCID.

**Table 1 ijms-23-06224-t001:** Research progress on neuroinflammation in VCID patients and VCID animal models.

	Subject	Immune Cells	Immune Mediators	Immune Signaling Pathways	References
Patient	Vascular cognitive impairment dementia	/	ACT, IL-6 ↑	/	[29]
Animal models	Bilateral Common Carotid Artery Stenosis (BCAS)	Activation of microglia and astrocytes	IL-1β, IL-6, TNF-α ↑	/	[110]
		Activation of microglia and astrocytes	ICAM-1, VCAM-1 ↑	/	[113]
		Oligodendrocyte precursor cell activation	MMP-9 ↑	/	[34]
		Axonal-glial integrity destruction	/	JAK/STAT ↑	[117]
		Activation of microglia and astrocytes; Demyelination	IL-1β, IL-6 ↑	NF-κB/STAT3 ↑	[124]
	2-Vessel Gradual Occlusion (2-VGO)	Activation of microglia and astrocytes	TNF-α, MCP-1 ↑	/	[111]
	Bilateral Common Carotid Artery Occlusion (BCAO)	Activation of microglia; Demyelination	IL-1β, IL-6, TNF-α ↑	C3-C3aR/ITGAM ↑	[86]
		Activation of microglia and astrocytes	IL-1β, IL-6, COX-2 ↑	TLR4/MyD88/p38/MAPK ↑	[112]
		/	ICAM-1, VCAM-1 ↑	/	[114]
		Activation of microglia and astrocytes	IL-6, TNF-α, MMP-2 ↑	/	[116]
		Activation of microglia	IL-6, TNF-α ↑	TLR4/MyD88/NF-κB ↑	[121]
	Asymmetric Common Carotid Artery surgery (ACAS)	/	IL-6 ↑ IL-4, IL-10 ↓	/	[109]
	Hyperhomocysteinemia (Hhcy)	Activation of microglia	MMP-2, MMP-9 ↑		[94]
		Activation of microglia; Activation of astrocytes, pseudopodia rupture and AQP4 channel decrease	IL-1β, IL-6, IL-12, TNF-α, MMP-2, MMP-9 ↑	/	[95]

## Data Availability

Not applicable.

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
