# Peer review of "Neuroinflammation in Vascular Cognitive Impairment and Dementia: Current Evidence, Advances, and Prospects"

_ijms, 2022, doi:10.3390/ijms23116224_

Round 1

Reviewer 1 Report

The work presents an interesting point of view on the associations between vascular cognitive impairment dementia and neuroinflammation. Authors show its heterogenous character. There are certain points which should be addressed: 

  1. In the introduction authors state: 

    "Vascular cognitive impairment dementia (VCID) is a severe cognitive dysfunction 31 syndrome caused by ischemia and hypoxia injury due to long-term chronic low perfusion 32 in the whole brain or local brain region. Further, it belongs to a group of heterogeneous 33 brain diseases caused by multiple factors, and its cognitive impairment is more severe 34 than other neurodegenerative diseases such as Alzheimer’ s disease" - authors should also acknowledge the possible vascular pathogenesis of syndromes typically associated with tauopathic pathology e.g. PSPS, CBS - Ref.

    Krzosek et al, Differential Diagnosis of Rare Subtypes of Progressive Supranuclear Palsy and PSP-Like Syndromes-Infrequent Manifestations of the Most Common Form of Atypical Parkinsonism. Front Aging Neurosci. 2022 Feb 9;14:804385. doi: 10.3389/fnagi.2022.804385. PMID: 35221993; PMCID: PMC8864174.

    Dunalska et al, The Significance of Vascular Pathogenesis in the Examination of Corticobasal Syndrome. Front Aging Neurosci. 2021 May 4;13:668614. doi: 10.3389/fnagi.2021.668614. PMID: 34017244; PMCID: PMC8129188

    2. A separate paragraph "Relationship between other tauopathic syndromes and VCID" should be added (as mentioned above).

    3. Authors could elaborate on how the association between neuroinflammation and vascular pathogenesis should interpreted as a point in modifying the course of the disease.

Author Response

We thank the reviewers and the editor for reviewing our paper and providing very valuable feedback. We have addressed each comment of reviewer 1 and provided our responses in this Word below. Thanks a lot.

Reviewer 2 Report

This review by Tian and colleagues is interesting. It highlights the complex pathophysiology of VCID and its relationship with multiple other pathologies.

A few minor comments:

There are figures and a table in the manuscript that are not referred to in the text. The 2 figures need a legend.

Line 54, please change 'ischemic' to 'ischemia'

Line 129, please change 'decreases' to 'decreased'

Line 244, please define NG2 cells, some reader might not know what they are.

Line 248, please change 'lead' to 'leading'

'central nervous system' is repeated 4 times between line 249 and 255. Please rephrase.

Author Response

We thank the reviewers and the editor for reviewing our paper and providing very valuable feedback. We have addressed each comment of reviewer 2 and provided our responses in this Word below. Thanks a lot.

Round 2

Reviewer 1 Report

Authors have significantly improved the manuscript, however there is one more issue, which could be beneficial.

Cognitive deterioration in atypical parkinsonian syndromes, as well as in other tauopathic syndromes may be additionally impacted by features typically associated with vascular risk factors as diabetes and prediabetes. This should be additionally highlighted in the section "6. Relationship between other tauopathic syndromes and VCID". REF.

1. The Rate of Decrease in Brain Perfusion in Progressive Supranuclear Palsy and Corticobasal Syndrome May Be Impacted by Glycemic Variability-A Pilot Study. Front Neurol. 2021 Nov 8;12:767480. doi: 10.3389/fneur.2021.767480. PMID: 34819913; PMCID: PMC8606811.

2. Common neurodegenerative pathways in obesity, diabetes, and Alzheimer's disease. Biochim Biophys Acta Mol Basis Dis. 2017 May;1863(5):1037-1045. doi: 10.1016/j.bbadis.2016.04.017. Epub 2016 May 6. PMID: 27156888; PMCID: PMC5344771.

3. Cognitive decline and dementia in diabetes mellitus: mechanisms and clinical implications. Nat Rev Endocrinol. 2018 Oct;14(10):591-604. doi: 10.1038/s41574-018-0048-7. PMID: 30022099; PMCID: PMC6397437.

Author Response

We thank the reviewers and the editor for reviewing our paper and providing very valuable feedback. And thanks a lot to reviewer for raising such important comments. We have addressed each comment of reviewer 1 and provided our responses in this Word below. 

Response to Reviewer 1 Comments

We thank the reviewers and the editor for reviewing our paper and providing very valuable feedback. We have addressed each comment of reviewer 1 below, and highlighted the changes in yellow in the revised manuscript. In the followings, the each valuable comment provided by reviewers will be highlighted in blue for easy identification.

Point 1: Cognitive deterioration in atypical parkinsonian syndromes, as well as in other tauopathic syndromes may be additionally impacted by features typically associated with vascular risk factors as diabetes and prediabetes. This should be additionally highlighted in the section "6. Relationship between other tauopathic syndromes and VCID". REF.

  1. The Rate of Decrease in Brain Perfusion in Progressive Supranuclear Palsy and Corticobasal Syndrome May Be Impacted by Glycemic Variability-A Pilot Study. Front Neurol. 2021 Nov 8;12:767480. doi: 10.3389/fneur.2021.767480. PMID: 34819913; PMCID: PMC8606811.

  1. Common neurodegenerative pathways in obesity, diabetes, and Alzheimer's disease. Biochim Biophys Acta Mol Basis Dis. 2017 May;1863(5):1037-1045. doi: 10.1016/j.bbadis.2016.04.017. Epub 2016 May 6. PMID: 27156888; PMCID: PMC5344771.

  1. Cognitive decline and dementia in diabetes mellitus: mechanisms and clinical implications. Nat Rev Endocrinol. 2018 Oct;14(10):591-604. doi: 10.1038/s41574-018-0048-7. PMID: 30022099; PMCID: PMC6397437.

Response 1: The reviewer raises an important point. We apologize that it was not added in the original submission. Thanks a lot to reviewer for pointing out these important references. We have now added this part in the revised manuscript (page 11, highlighted in blue). The added text reads as follows:

Page 11, Line 447-460: Current studies have shown that vascular risk factors have become an important path-ogenesis of tau disease. On the one hand, hypertension or cerebral ischemia, oxidative stress and other vascular risk factors will induce pathological changes of tau, resulting in motor disorders[1,2]. For example, The P301L-tau Tg mouse model is one of the best characterized mouse models of tauopathy, and the motor disturbances developed by them have been correlated with the presence of hyperphosphorylated tau[3-5]. On the other hand, vascular risk factors can lead to cognitive dysfunction by inducing neuro-logical damage. For example, diabetes has become the main risk factor for PSP, and the decrease of cerebral perfusion in PSP may be regulated by blood glucose variability, thereby inducing the occurrence and development of cognitive dysfunction[6,7]. In addition, diabetes can also be as an important risk factor to participate in the cognitive decline process of other neurodegenerative diseases via mechanisms that involve cere-brovascular pathologies, such as AD[8-10].

References

  1. Dunalska, A.; Pikul, J.; Schok, K.; Wiejak, K.A.; Alster, P. The Significance of Vascular Pathogenesis in the Examination of Corticobasal Syndrome. Front Aging Neurosci 2021, 13, 668614, doi:10.3389/fnagi.2021.668614.
  2. Mulroy, E.; Jaunmuktane, Z.; Balint, B.; Erro, R.; Latorre, A.; Bhatia, K.P. Some New and Unexpected Tauopathies in Movement Disorders. Mov Disord Clin Pract 2020, 7, 616-626, doi:10.1002/mdc3.12995.
  3. Díaz-Ruiz, C.; Wang, J.; Ksiezak-Reding, H.; Ho, L.; Qian, X.; Humala, N.; Thomas, S.; Martínez-Martín, P.; Pasinetti, G.M. Role of Hypertension in Aggravating Abeta Neuropathology of AD Type and Tau-Mediated Motor Impairment. Cardiovasc Psychiatry Neurol 2009, 2009, 107286, doi:10.1155/2009/107286.
  4. Le Corre, S.; Klafki, H.W.; Plesnila, N.; Hübinger, G.; Obermeier, A.; Sahagún, H.; Monse, B.; Seneci, P.; Lewis, J.; Eriksen, J.; et al. An inhibitor of tau hyperphosphorylation prevents severe motor impairments in tau transgenic mice. Proc Natl Acad Sci U S A 2006, 103, 9673-9678, doi:10.1073/pnas.0602913103.
  5. Berger, Z.; Roder, H.; Hanna, A.; Carlson, A.; Rangachari, V.; Yue, M.; Wszolek, Z.; Ashe, K.; Knight, J.; Dickson, D.; et al. Accumulation of pathological tau species and memory loss in a conditional model of tauopathy. J Neurosci 2007, 27, 3650-3662, doi:10.1523/jneurosci.0587-07.2007.
  6. Kwasny, M.J.; Oleske, D.M.; Zamudio, J.; Diegidio, R.; Höglinger, G.U. Clinical Features Observed in General Practice Associated With the Subsequent Diagnosis of Progressive Supranuclear Palsy. Front Neurol 2021, 12, 637176, doi:10.3389/fneur.2021.637176.
  7. Alster, P.; Dunalska, A.; Migda, B.; Madetko, N.; Królicki, L. The Rate of Decrease in Brain Perfusion in Progressive Supranuclear Palsy and Corticobasal Syndrome May Be Impacted by Glycemic Variability-A Pilot Study. Front Neurol 2021, 12, 767480, doi:10.3389/fneur.2021.767480.
  8. Biessels, G.J.; Despa, F. Cognitive decline and dementia in diabetes mellitus: mechanisms and clinical implications. Nat Rev Endocrinol 2018, 14, 591-604, doi:10.1038/s41574-018-0048-7.
  9. Pugazhenthi, S.; Qin, L.; Reddy, P.H. Common neurodegenerative pathways in obesity, diabetes, and Alzheimer's disease. Biochim Biophys Acta Mol Basis Dis 2017, 1863, 1037-1045, doi:10.1016/j.bbadis.2016.04.017.
  10. Devi, L.; Alldred, M.J.; Ginsberg, S.D.; Ohno, M. Mechanisms underlying insulin deficiency-induced acceleration of β-amyloidosis in a mouse model of Alzheimer's disease. PLoS One 2012, 7, e32792, doi:10.1371/journal.pone.0032792.

Round 3

Reviewer 1 Report

I do not have further comments.